# Leak Detection in Water Pipes Using Submersible Optical Optic-Based Pressure Sensor

**DOI:** 10.3390/s18124192

**Published:** 2018-11-30

**Authors:** Leslie Wong, Ravin Deo, Suranji Rathnayaka, Benjamin Shannon, Chunshun Zhang, Wing Kong Chiu, Jayantha Kodikara, Hera Widyastuti

**Affiliations:** 1Department of Civil Engineering, Monash University, Clayton Campus, Victoria 3800, Australia; ravin.deo@monash.edu (R.D.); Suranj.rathnayaka@monash.edu (S.R.); benjamin.shannon@monash.edu (B.S.); ivan.zhang@monash.edu (C.Z.); jayantha.kodikara@monash.edu (J.K.); 2Department of Mechanical and Aerospace Engineering, Monash University, Clayton Campus, Victoria 3800, Australia; wing.kong.chiu@monash.edu; 3Institute Teknologi Sepuluh Nopember, Jl Arief Rahman Hakim, Surabaya 60111, Indonesia; hera.widyastuti@yahoo.co.uk

**Keywords:** optical fibre sensors, leak detection, sensors, nondestructive technique, pipeline

## Abstract

Leakage is undesirable in water distribution networks, as leaky pipes are financially costly both to water utilities and consumers. The ability to detect, locate, and quantify leaks can significantly improve the service delivered. Optical fibre sensors (OFS) have previously demonstrated their capabilities in performing real-time and continuous monitoring of pipe strength leak detection. However, the challenge remains due to the high labour cost and time-consuming process for the installation of optical fibre sensors to existing buried pipelines. The aim of this paper is to evaluate the feasibility of a submersible optical fibre-based pressure sensor that can be deployed without rigid bonding to the pipeline. This paper presents a set of experiments conducted using the proposed sensing strategy for leak detection. The calibrated optical fibre device was used to monitor the internal water pressure in a pipe with simultaneous verification from a pressure gauge. Two different pressure-based leak detection methods were explored. These leak detection methods were based on hydrostatic and pressure transient responses of the optical fibre pressure sensor. Experimental results aided in evaluating the functionality, reliability, and robustness of the submersible optical fibre pressure sensor.

## 1. Introduction

Optical fibre sensors (OFS), notably the fibre Bragg gratings (FBGs) and distributed optical fibre sensors (DOFS) have gained a reputation over the years and demonstrated their potential applications in pipeline structural health monitoring applications [1,2,3,4]. 

FBGs are well-known for point sensing. An interrogator is used to monitor the Bragg wavelength shift of intrinsic sensing elements, which have been inscribed into the core of an optical fibre. They are commonly used for monitoring localised physical quantities, such as strain, temperature, and pressure [5,6]. The descriptions of FBGs and their applications are widely reviewed and discussed by the authors of [7,8,9,10]. FBGs can also be integrated into different structures to develop novel optical fibre-based sensors. FBGs can be multiplexed to form a series of FBGs along a fibre cable, which makes it possible to perform as a quasi-distributed measurement. However, this can become very costly for constructing a high dense array of FBGs.

DOFS, namely distributed strain and temperature sensing (DSTS), distributed temperature sensing (DTS), and distributed acoustic sensing (DAS), offer distributed and continuous monitoring at every single point along a sensing cable. DOFS technologies have been developed based on measuring the intrinsic backscattering, including Rayleigh, Brillouin, and Raman scattering [11,12], based on optical time domain reflectometry (OTDR) and optical frequency domain reflectometry (OFDR). DSTS is commonly used for strain sensing by installing and mounting directly to the pipes [13]. The applications of DOFS, in particular of strains measurement for buried pipelines health monitoring, are reviewed in References [1,14,15]. DSTS is capable of detecting, locating, and monitoring ground movement [16] and the bending and buckling effects of a pipe with three optical cables (strategically aligned 120° apart) installed parallel along a pipe’s circumference. The optical sensors can also be instrumented helically to provide circumferential information [17,18] along the pipe. In some laboratory and field trial, DSTS was used for crack monitoring [19], investigating the change in pipe wall thickness [18], and to determine the localised stiffness changes on a pressurised out-of-roundness flexible polyvinyl chloride (PVC) pipe [17]. Some progress has also been made recently in improving both the high spatial resolution and large dynamic range of DSTS technology, in particular with Rayleigh-based optical frequency domain reflectometry (OFDR) techniques [20]. Wong et. al. [21,22] showed the potential of using DSTS to monitor the dynamic response of a small diameter pipe when subjected to pressure transients. Optical fibre sensors can also be used to detect and monitor the crack growth due to fatigue transient loading [23]. Schenato et al. [24] developed DSTS-based single-point pressure and temperature sensing for riverbanks monitoring using optical backscattering reflectometry (OBR) from Luna Technologies with subcentimetre resolution and monitoring within seconds. 

Eisler et al. [25] proposed using distributed temperature sensing (DTS) to detect leakage, using optical fibre cables that were embedded in soil, close to the pipeline. Their results [25] showed that embedded fibre sensors can reliably detect and localise movement in the soil that can potentially endanger the pipe. This was found to be quicker and easier than bonding the sensors onto the pipe [26]. Higgins and Paulson [27] performed a field trial using a distributed acoustic sensing (DAS) system to monitor the acoustic signal due to the breaking wires in a concrete pipe. Their result showed that optical fibre sensors are more stringent than hydrophone arrays, as the fibre cable was in close proximity to the wire break. A large-scale laboratory trial conducted by Wu et al. [28] demonstrated that by installing a phase-sensitive-based DAS optical fibre cable on the inner pipe wall, a leak can be detected. Liu [29] and Miah [30] also provide a thorough review on distributed optical fibre sensors for vibration detection and geophysical applications in pipes. It is also noted that the DAS technique cannot offer high spatial resolution.

It is noted that there are three most common fibre orientations used on pipelines, which are identified as (1) mounted or embedded in circumferential (hoop), (2) longitudinal (axial), and (3) helical (spiral) directions along the pipeline. These three orientations are illustrated in Figure 1.

Almost all of the optical fibre sensors’ deployment methods would require the optical fibre to be rigidly bonded to the pipe or embedded in the soil. These optical fibre deployment methods are feasible for new and smart pipelines (sensors-based self-monitoring pipeline). A challenge remains in the use of fibre sensors for the assessment of existing buried and old pipes. The present study presents a cost-effective sensor deployment strategy that can be utilised in buried and old pipes. The approach presented facilitates the application of optical fibre sensors when the defect location is not well defined along the pipes—a common practical problem. For this purpose, an “attachment-free” fibre deployment method is explored and examined. An optical fibre-based pressure sensor, based on the “attachment-free” concept, was designed and constructed. Investigation of the use of this sensor for the quantification of pipe leakage was performed. Based on observation and results, the feasibility of the design concept for indirectly quantifying pipe leak rates through the monitoring of the pressure transients are discussed.

## 2. Design of Submersible Optical Fibre-Based Pressure Sensor

The submersible optical fibre-based pressure sensor developed in this work is a customised optical fibre-based prototype pressure transducer constructed at Monash University, Australia. The proposed pressure sensor can be deployed into the water pipes via a hydrant. The advantage of this deployment method is that the location of the sensors can be adjusted based on the needs, whereas the commercialised pressure transducer can only be installed on the location of the hydrant.

A schematic of a single unit customised sensor is given in Figure 2a. A single strand of single mode fibre (SMF28e) was used for the testing. PVC was selected as the casing material for the first prototype sensor as it is easy to obtained and cheap in price. The optical fibre sensor was bonded on the internal surface of a cylindrical 100 mm-long PVC tube (inner diameter = 25 mm, thickness = 1 mm) with 5 min Araldite, as shown in Figure 2b. Both ends of the PVC tube were appropriately sealed with end caps (with optical feedthrough). The total effective sensing length of the optical fibre was 50 mm. The loose fibre ends were protected with a rigid rubber tube (diameter = 4 mm). All the connection points were sealed with water-proof silicone. The pressure fluctuations will be determined from the changes in the surface strain of the host structure (PVC tube). Figure 2c shows a concept design used in the experiment. It is noted that this design survived the tests described in the paper, attesting to its robustness.

## 3. Calibration Test

The optical fibre from the device in Figure 2 was connected to the optical distributed sensor interrogator (ODiSI-B series) from LUNA Technologies. The ODiSI-B operates based on the Rayleigh optical frequency domain reflectometry (OFDR) coupled with swept-wavelength interferometry (SWI). The essential specifications of this interrogator are listed in Table 1. The OFDR system was used for testing purposes on our devices as it offers both high spatial and temporal resolution. The system can measure up to a maximum length of 10 m at 100 Hz. The system has been previously used by Wong et al. [31,32] for monitoring composite and cement structures.

The schematic drawing of the experimental setup is presented in Figure 3a. Two 1 m u-PVC pipes with an inner diameter of 100 mm and a thickness of 3 mm were prepared and assembled with a T-joint (see Figure 3b) to make a total pipe length of 2 m for the experiment. A hole of 34 mm in diameter was drilled at the centre of three end caps. The prototype sensor, as described in Section 2, was installed onto one of the end caps and properly sealed using waterproof silicone. When the optical fibre-based sensor was placed in the PVC pipe, its location was 1.2 m away from the end cap (location of insertion). A water inlet and a pressure transducer were connected to another end of the PVC pipe (on the end cap). A ball valve was connected to the last end cap and installed on the T-joint. This ball valve served two purposes, (1) to simulate a through-wall leak in the pipe and to also (2) allow to control the leak rate. The experimental setup is shown in Figure 3b.

The optical fibre-based pressure sensor was first calibrated with the PVC pipe filled with water under a quasistatic condition. The ball valve was left open during the filling process to allow the air within the pipe to be exhausted. After the pipe was filled, the ball valve was closed. It is noted that there was a baseline offset due to the presence of water, as the sensor is sensitive to change in the surrounding temperature. As there was no thermal couple installed in our prototype sensor to compensate for the temperature change, it was proposed to reset the baseline before each experiment. Three pressurising and depressurising tests were conducted on the pipe by connecting and disconnecting the inlet of the pipe to the water supply with a known pressure (500 kPa). The optical fibre sensor was calibrated against the pressure transducer. The measurements were recorded at a sampling rate of 100 Hz.

ODiSI was used to monitor the strain along the entire fibre sensors. The strain measurement along the fibre is plotted in the contours plot in Figure 4a. As shown in Figure 4a, approximately 50 mm of fibre-bonded region shows a compressive strain (red) when subjected to pressure changes. The strain measured at 80 s is plotted in Figure 4b. The measured strain is not uniform across the bonded fibre. The maximum strain was measured at 1510 mm. Rather than applying averaging on the strain measured, the point with the maximum absolute strain value measured was chosen as the point of monitoring, as it provides the highest sensitivity. The strain measurements and corresponding pressure results obtained using the optical fibre sensor at the point (with maximum strain) over time are plotted in Figure 4c.

Figure 4a–c shows the negative strain measured (compression) when the sensor was subjected to an increment of water pressure. The negative strain indicates that the water pressure was compressing the entire structure (tube with fibre sensors) as a whole and causing a contraction effect on the bonded fibre sensor. The pressure measured was then plotted against the strain output from the optical fibre device (see Figure 4d). The strong linear relationship between the optical fibre device output and the pressure within the pipe is evident. According to the specification of ODiSI, the minimum detectable change in strain (strain resolution) is 1 με and a repeatability of ±5 με, which corresponds to a pressure resolution of this prototype of 4.46 kPa and a repeatability of ±22.3 kPa. The sensitivity of this prototype sensor is lower than that of the traditional pressure transducer (1 kPa). The sensitivity and accuracy of the device can be further improved by fine-tuning the casing using a different geometry (thinner casing material) or material selection. Nevertheless, for this first prototype, it is still sufficient for the desired applications with less than 4% error, given the maximum monitored water pressure of 600 kPa.

As the fibre sensors are sensitive to both strain and temperature, the unbonded fibre was also monitored throughout the test so that the change in temperature can be compensated. By monitoring the unbonded region of the fibre, no significant change in temperature was measured throughout the test, as shown in Figure 4a,b. Nevertheless, it is a common practice to have thermal compensation (using thermal couple or Raman-based distributed temperature sensing) set up in parallel with the strain-based fibre system for practical applications.

Following calibration, the optical fibre device was verified under dynamic loading conditions. To achieve this test condition, a series of varying operating conditions (involving the connecting and disconnecting of the water supply) were used to generate the dynamic conditions. This allowed a validation of the dynamic response of the submersible optical fibre-based pressure sensor to be conducted. Results obtained from both pressure measuring systems are shown in Figure 5. These results show excellent agreement between the magnitude and the dynamic responses obtained from the optical fibre device and the pressure transducer.

## 4. Potential Applications of the Submersible Optical Fibre-Based Pressure Sensors

The submersible optical fibre-based pressure sensor demonstrated in Section 3 was used in the following experiments. The experiments involved using the optical fibre device to:(i)Determine the operating condition of the pipe (from both steady and unsteady pressure occurring within the pipe section);(ii)Perform leak detection when the pipe was subjected to hydrostatic pressure;(iii)Perform leak detection when the pipe was subjected to pressure transients;(iv)And detect the presence of pipe fracture (pipe burst).

Results for each of the experiments are summarised in the relevant sections below.

### 4.1. Monitoring Pipeline Operating Conditions

A series of changes in operating conditions were carried out using the same experimental setup, shown in Figure 3. Both a submersible optical fibre-based pressure sensor and pressure transducer were used to measure the water pressure at a sampling rate of 100 Hz. The pressure profiles obtained from both sensors are reported in Figure 6. The results presented in Figure 6 show the agreement between the two measurement sensors.

Figure 7a–c shows detailed views of different operation conditions when the pipe was subjected to Actions 1–6. Due to the sudden changes in operating conditions, transient events were induced. The pressure fluctuations are evident from the measurements obtained from both the conventional pressure transducer and optical fibre-based pressure sensor shown in Figure 7a–c. Figure 7a shows the pressure profile when the valve connecting the pipe and the water supply (operating pressure = 450 kPa) was turned on (Action 1). A pressure transient can be observed clearly in Figure 7a and this transient event dampens out over time (approximately 8 s). As the conventional pressure transducer is closer to the water supply, the overshoot of the measurement obtained from the pressure transducer is higher (maximum pressure at 530 kPa) than the measurement obtained from the optical fibre-based pressure sensor (maximum pressure at 500 kPa). However, both measurements yielded at 450 kPa when the pressure in the pipe reached equilibrium after 8 s.

Next, the ball valve previously installed on the tee-junction (see Figure 3) was turned on and shut off rapidly (Action 2) to induce a transient event. Before the transient event dampened out, the ball valve was turned on a quarter (approximately 45°) for another 2 s (Action 3) to simulate sudden leakage along the pipe. The measured response is presented in Figure 7b. During the leak event, the pressure dropped to 400 kPa. The ball valve was then shut off after 2 s. The ball valve was turned on and off rapidly (Action 4) again. The response due to Action 4 is reported in Figure 7b. The action of opening and closing the ball valve caused some pressure fluctuations and they are clearly shown in Figure 7b. This shows that the optical fibre device is very sensitive to pressure changes and is capable of performing real-time and continuous monitoring when the optical fibre device is connected to the interrogator with sufficient spatial and temporal resolution. 

Before the transient pressure dampened out due to the fast closing of the valve (Action 4), the ball valve was turned back on to approximately 45° and was held for 15 s (Action 5). In Figure 7c, it is shown that when this simulated leak event occurred, the static pressure within the pipe decreased from 450 kPa to 400 kPa. There are higher frequency components in the pressure measurement obtained from the optical fibre device due to Action 5, whereas the commercialised pressure transducer was unable to detect the small fluctuations. The fluctuations can be attributed to the location of the optical fibre-based sensor, as it is closer to the leak location than the pressure transducer. This result shows that the position of the sensor is vital to pinpointing the leak. After 15 s (Action 5), the ball valve was shut rapidly (Action 6). There are some pressure fluctuations due to the rapid closure of the ball valve. The pressurised water supply was then disconnected (Action 7). There are leaks in the connections, which caused the hydrostatic pressure to drop after disconnecting the water supply, see Figure 7d. Finally, the excessive water pressure was released by turning the valve at the water outlet (Action 8). The results obtained thus far have shown the capability and reliability of the submersible optical fibre-based pressure sensor to monitor both steady and unsteady pressure. 

### 4.2. Pressure-Based Leak Detection

The two common pressure-based leak detection methods are the hydrostatic-based and transient-based leak detection methods. In the following section, the submersible optical fibre-based pressure sensor was used to demonstrate the capability of leak detection under both static and dynamic water pressure fluctuations by analysing the signature of the monitored pressure.

#### 4.2.1. Static Leak Detection Technique (Hydrostatic Testing)

Hydrostatic testing is a common leak detection method in the field. A pipe section will be subjected to a higher static pressure for a long period of time (generally eight hours). The high pressure will accentuate all the features on the pipe and thus enhance the process of structural health assessment. Any abnormal pressure drop will indicate the presence of leakages [33]. The previous experimental setup (see Figure 3) was used to conduct a small-scale hydrostatic testing. The pipe was first connected to the water supply (reservoir with 500 kPa) with the ball valve at the tee-junction fully closed (at 0°). When the pressure reached the target static pressure (500 kPa), the ball valve on the tee-junction was opened partially (10° in Figure 8a) to simulate leakage. The handle (of the ball valve) was held for approximately 15 s at the same position to simulate leakage and then closed (0°). The testing procedure was repeated, opening and closing the valve to 20°, 30°, and 40° when the static pressure within the test pipe attained 500 kPa. Figure 8b shows the leakage scenario when the ball valve was set to 30°. The deployed optical fibre-based pressure sensor was used to measure the pressure profile of this series of experiments at 100 Hz and the results for pressure loss and leak rate are presented in Figure 9. The results in Figure 9 show that the deployed optical fibre device was capable of determining the pressure drop due to leakage. This represents a potential for this “attachment-free” device to be deployed where a permanently mounted pressure transducer is not feasible. This “attachment-free” optical fibre device can be seen as an effective and economical solution for leakage detection.

When the ball valve was opened partially, the internal water pressure dropped, as shown in Figure 9. The pressure drop, ΔP, was then calculated by finding the difference between the mean pressure over 10 s with leakage and the mean water pressure over 10 s with no leak at 500 kPa constant pressure, excluding the small pressure fluctuations. The relationship between the pressure drop and leak rate against the ball valve handle orientation (in terms of degrees) is presented in Figure 10a. Since the ball valve was a quarter-turn valve (turning 90° to fully close or open), the leak rate and pressure drop does not increase linearly with the turn angle (opening the water outlet is not a linear increase with turning angle). This behaviour is reflected in Figure 10a. A linear relationship between the pressure drops and leak rate was found and can be seen in Figure 10b, which shows that the leak rate can be quantified by monitoring the pressure profile obtained in the pipe. With a better analysis of the pressure profile, it is possible to relate or even quantify the leak rate to pressure drops when the pipe is subjected to hydrostatic testing in the field. It is also noted that the actual leaks can be very complex and depend on many factors, such as crack morphology; therefore, further research is needed to simulate a field scenario (a more complex leak situation including multiple demands and discharges).

#### 4.2.2. Transient-Based Leak Detection

In the last two decades, many fluid transient-based leak detection techniques have been developed [34,35,36,37]. According to Ramos et al. [38] and Wang et al. [39], the time history of the pressure head, h, can be formulated in terms of an exponential law:(1)$$h=me^{-\xi t},$$
where ‘*m*’ is the initial pressure maxima, ‘*ξ*’ is decay or damping coefficient, and ‘*t*’ is time. Wang et al. [40] have also been able to quantify the leak size and locate the leak position based on the leak-induced damping. Gong et al. [41] have also performed in-pipe pressure sensing for leak detection using a time-domain analysis.

In this section, the submersible optical fibre pressure sensor was used to demonstrate transient-based leak detection and frequency-domain analysis using the setup as shown in Figure 3. In this test, 5 end caps were prepared by drilling a circular hole (damage) at the centre on four of the end caps. The diameters of the drilled hole on each end cap were 2.02 mm, 4.44 mm, 7.18 mm, and 9.52 mm, as shown in Figure 11. The leak size is defined as the diameters of the holes, which are approximately 3.3 mm^2^, 15.5 mm^2^, 40.5 mm^2^, and 71.4 mm^2^, respectively. Initially, the end cap without any damage (reference end cap) was installed on the tee-junction (replacing the ball valve) to simulate a no leak situation. By turning on the valve that connects the pipe to the water supply with 500 kPa (Action 1), a transient event is generated and propagated into the pipe. The transient event was monitored using the optical fibre device at a data acquisition rate of 100 Hz. The experiment was then repeated with the four different (damaged) end caps (see Figure 11) one at a time. Normalised pressure profiles for each of these transient cases are reported in Figure 12.

For the pipe without leakage, the transient event dampened out in approximately 6 s (see Figure 12). The initial damping coefficient is caused by several factors [36,40], including the geometry of the pipe, friction, and the existing fittings within the pipe network. The presence of a severe leak can lead to an increase in the damping coefficient or damping rate. It is evident that the transient event dampens out faster (less than 4 s) due to the presence of leakage, as shown in Figure 12. The rapid decaying pressure transient agrees with the behaviour of the transient signal in the leak system reported by Colombo et. al. [36]. The increment of transient damping can be studied to achieve detection and is also known as the key premise of transient-based leak detection.

The damping ratio was then calculated for each of the transient events presented in Figure 12 through a least-squares nonlinear fitting procedure. The damping ratios for different leak size are reported in Figure 13. In Figure 13, the leak size was normalised with the cross-sectional area of the pipe (diameter of 100 mm). This result shows that it is possible to quantify the leak size by identifying the leak-induced damping coefficient. 

The data analysis method for this ‘attachment-free’ optical fibre sensor deployment method is not established yet. As a trial, spectral analysis (frequency domain analysis) was performed on the results presented in Figure 12 to demonstrate the effect of the damping. The results are presented in spectrograms, as shown in Figure 14a–e. With the increment of leak size, the frequency of the pressure transient shifted downward (reduced) is evident in Figure 14a–e. The damped frequencies obtained from each leak size, including the reference (no leak case), are plotted in Figure 14f. The leak-induced damping causes the frequency shift in the pressure transient. Further research is required to predict and quantify the leak sizes in a real field application.

### 4.3. Detection of Pipe Burst/Fracture

Pipe burst is a very common failure mode for pipes in the field when defects in plastic pipes, or severely corroded pits and patches in cast iron pipes are subjected to excessive (internal or external) loadings [42,43]. With the “attachment-free” optical fibre device, it is possible to detect pipe bursts along the pipeline. In one of the pressurised experiments conducted, the end cap burst and caused the pressure to drop instantly. This situation somewhat simulates an actual pipe burst event in the field. The results obtained by both pressure-measuring systems during this event are reported in Figure 15. Both the commercialised pressure transducer and the submersible optical fibre pressure sensor did not suffer any damage, demonstrating their robustness under such conditions. 

The results thus far presented in this paper are only based on one (point) sensor with a pressure sensitive length of 50 mm and acquisition rate of 100 Hz. To the best of the authors’ knowledge, at this stage, there is no long-range distributed optical technique that offers both high spatial and temporal resolution simultaneously. A lower spatial resolution interrogator will measure a lower strain along the pressure sensitive length, which reduces the sensitivity of the developed device. However, discrete sensing, like FBG technology, may be adopted for practical field deployment. With FBG technology, the same sensor array can also easily be multiplexed to form an array of sensors that can be deployed as quasi-distributed sensing in the future. With an increment of sensing points, it can potentially improve the determination of the leaks and anomalies along the pipelines. Multiple (optical fibre) sensors with different monitoring purposes (DTS and DAS) can also be installed into the submersible pressure sensors so that a data fusion can be performed and hence reduce the chance of a false positive scenario [44,45]. 

## 5. Conclusions and Future Plan

It is always challenging to install optical fibre sensors to existing buried water pipelines. Almost all of the deployment methods of optical fibre sensors require the optical fibres to be mounted to the surface of the pipe to provide useful information for the condition assessment of the pipe and leak detection. Any damage to the optical fibre sensors would require digging up to access to the fibre sensors. An “attachment-free” deployment method was proposed and examined experimentally in this paper. A submersible optical fibre-based pressure sensor (prototype) was built and tested under laboratory conditions. The optical fibre device was first calibrated and successfully demonstrated the capability to monitor both the static and dynamic (transient pressure) water pressure. This paper also reported the application of the optical fibre device to perform a pressure-based leak detection method and detection of pipe burst. If the proposed sensors are damaged, a replacement can be done by extracting the whole sensors back from the insertion point. 

Given that ODiSI has sufficient spatial resolution to measure strain, as required for the working mechanism of the developed device, it was used to monitor the strain (or pressure) of the submersible optical fibre-based pressure sensor. However, the inner working mechanism of the sensor does not rely on this specific optical technique to measure the strain and temperature, as it might not be practical to be applied in the field due to the short monitoring range (up to 20 m for ODiSI). To the best of the authors’ knowledge, the technology able to offer both high spatial and temporal resolution over long distance is yet to be developed. The FBG technique will be more preferable for practical field application of the proposed sensing device at this stage. By adopting the FBG technology, this submersible sensor can easily be multiplexed so that the monitoring range could extend to over several kilometres. The DTS and DAS technologies can potentially be infused in the submersible sensor to offer multiparameter monitoring devices.

In the presented work, PVC was chosen as the material for building our first prototype due to the ease of fabrication and cost. However, it is also well-known that PVC is nonlinear, can exhibit hysteresis and creep, and is a strongly temperature-dependent material. Further development and material selection of the sensor is ongoing to improve the accuracy and sensitivity of the sensor. This can be achieved by varying the geometry of the sensor and material of the host structure (where the fibre is attached). In future work, different types of casing material and short and long-term material effects will also be studied to extend the application for a commercial deployment. Since optical fibre sensors can be multiplexed, it is possible to duplicate the submersible optical fibre-based pressure sensor along the pipe. With more pressure-monitoring points, potential improvement of the location of leak/s and quantification of leak rates can be monitored and ultimately, better prediction of the remaining servicing life of the pipe can be achieved.

## Figures and Tables

**Figure 1 sensors-18-04192-f001:**
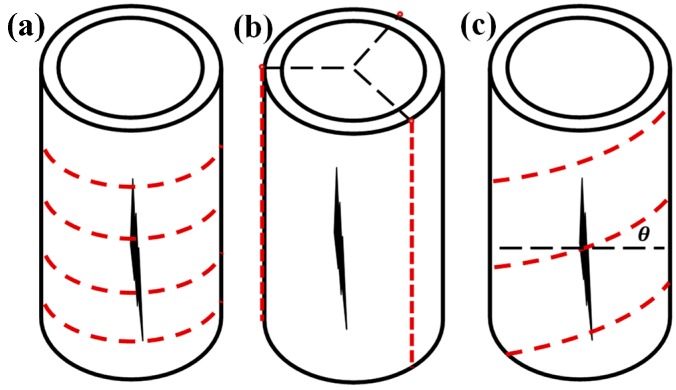
Common distributed optical fibre sensor orientation, (**a**) hoop direction, (**b**) axial direction, and (**c**) helical direction.

**Figure 2 sensors-18-04192-f002:**
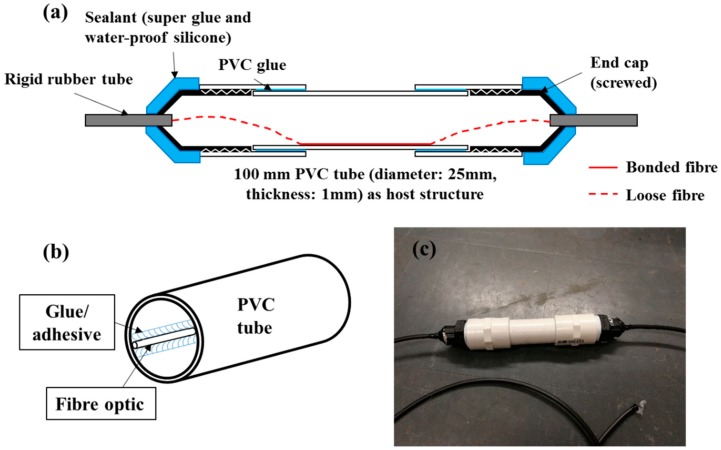
(**a**) Schematic drawing of the prototype submersible optical fibre pressure sensor, (**b**) depiction of interior of the PVC tube, and (**c**) depiction of the actual sensor.

**Figure 3 sensors-18-04192-f003:**
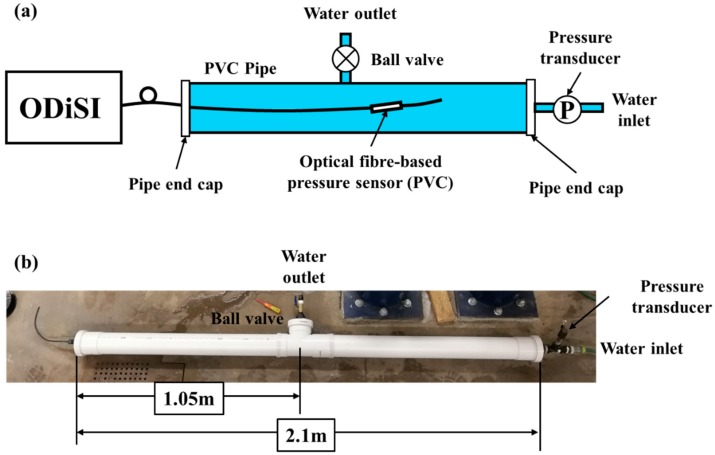
(**a**) Schematic of the experimental setup and (**b**) the actual experimental setup.

**Figure 4 sensors-18-04192-f004:**
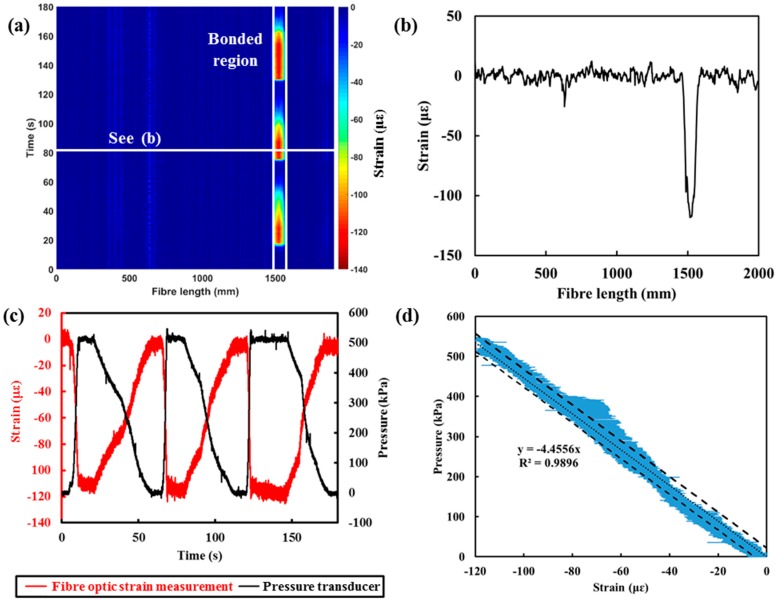
(**a**) Strain measurement (contour plot) obtained by ODiSI, (**b**) strain measurement along the entire fibre sensors at 80 s, (**c**) comparison between the water pressure measured by the pressure transducer and the strain with the submersible optical fibre-based pressure sensor, and (**d**) correlation between the pressure and strain measured.

**Figure 5 sensors-18-04192-f005:**
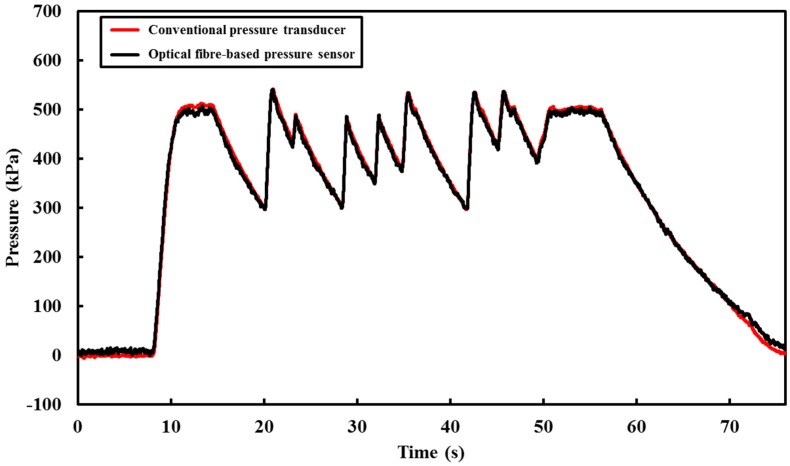
Comparison between the optical fibre-based pressure sensor and the traditional pressure transducer.

**Figure 6 sensors-18-04192-f006:**
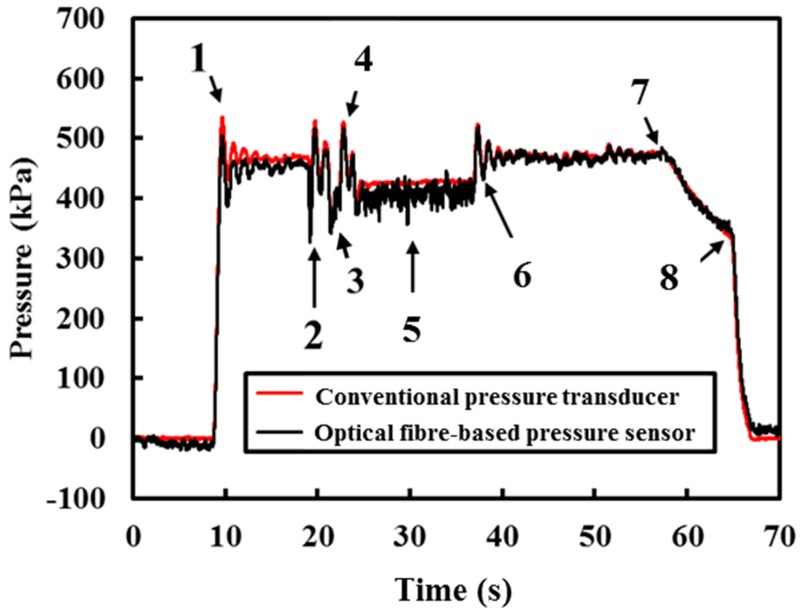
Comparison between the conventional pressure transducer and optical fibre-based sensor when the pipe was subjected to various changes (Actions 1–8) in pipe operating condition.

**Figure 7 sensors-18-04192-f007:**
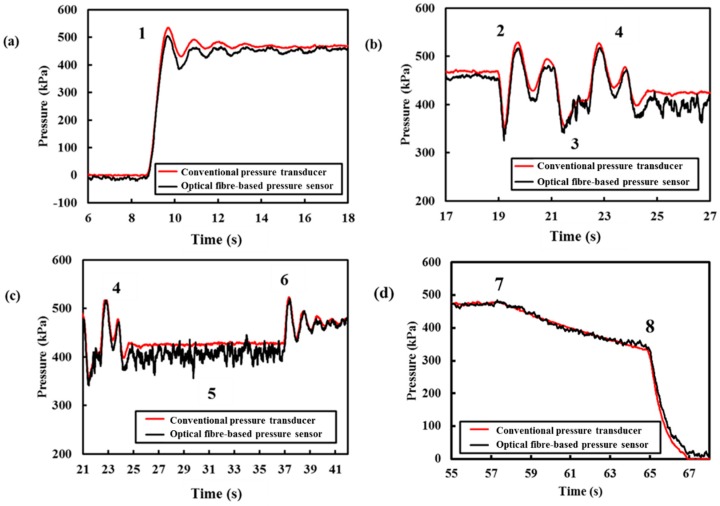
Zoomed in on Figure 6 at (**a**) Action 1, (**b**) Actions 2–4, (**c**) Actions 4–6, and (**d**) Actions 7 and 8.

**Figure 8 sensors-18-04192-f008:**
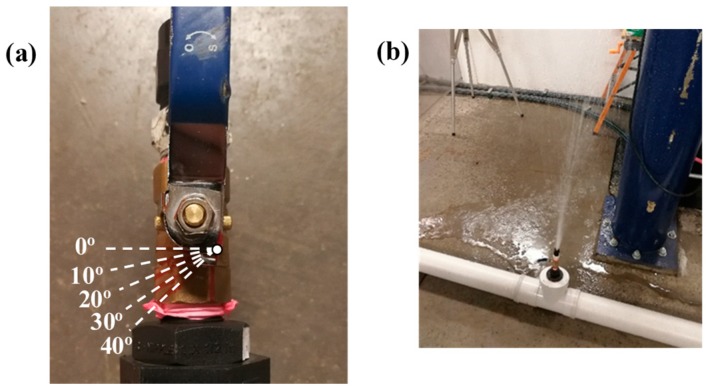
(**a**) Ball valve with angle labelled to simulate different leak rate and (**b**) leaking pipe with the handle of the ball valve set at 30°.

**Figure 9 sensors-18-04192-f009:**
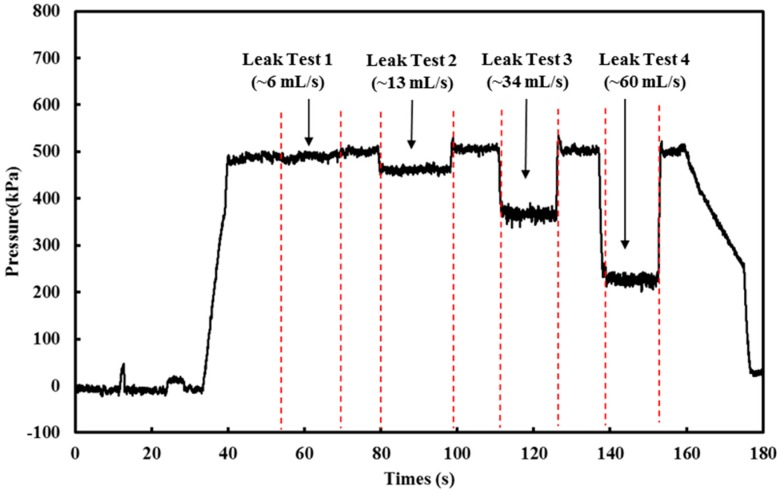
Pressure evolution for the hydrostatic testing (measured by the optical fibre-based pressure sensor).

**Figure 10 sensors-18-04192-f010:**
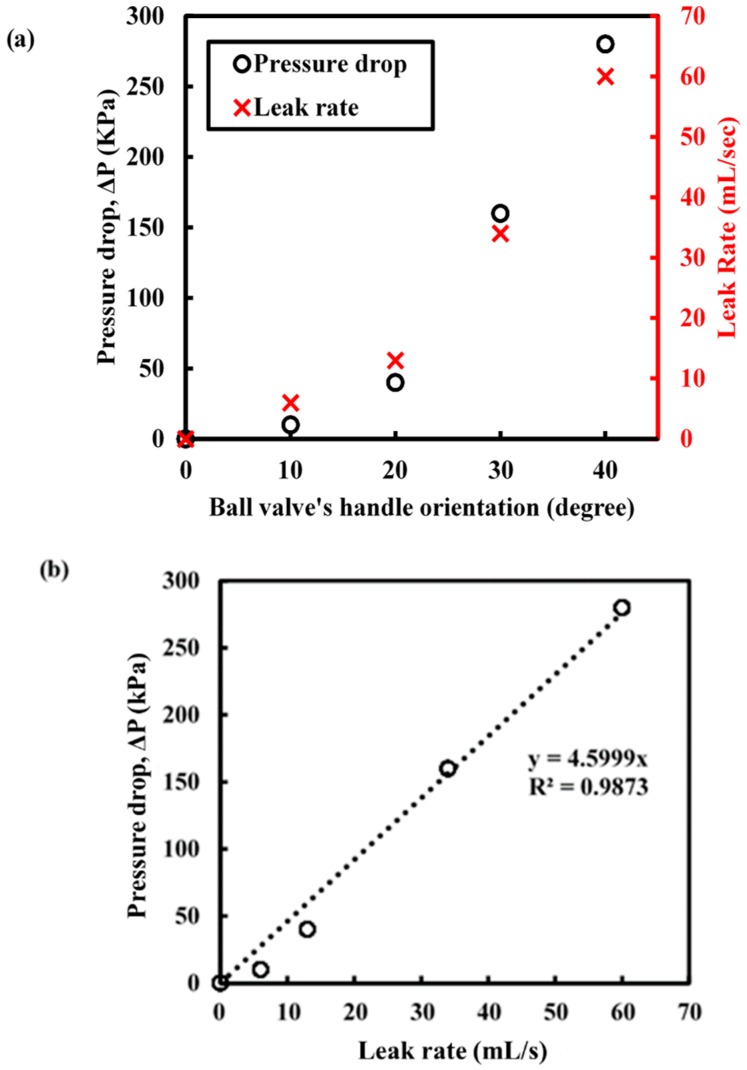
(**a**) Effect of pressure drop (measured using the optical fibre device) and leak rate due to different leakage simulation scenario and (**b**) the correlation between pressure drops against leak rate.

**Figure 11 sensors-18-04192-f011:**
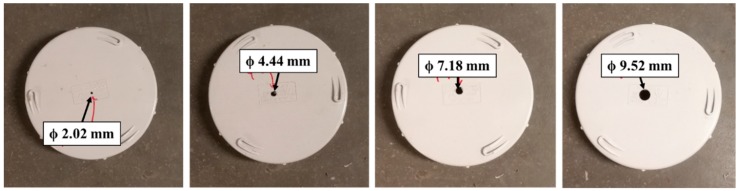
End caps with a different drilled diameter at the centre.

**Figure 12 sensors-18-04192-f012:**
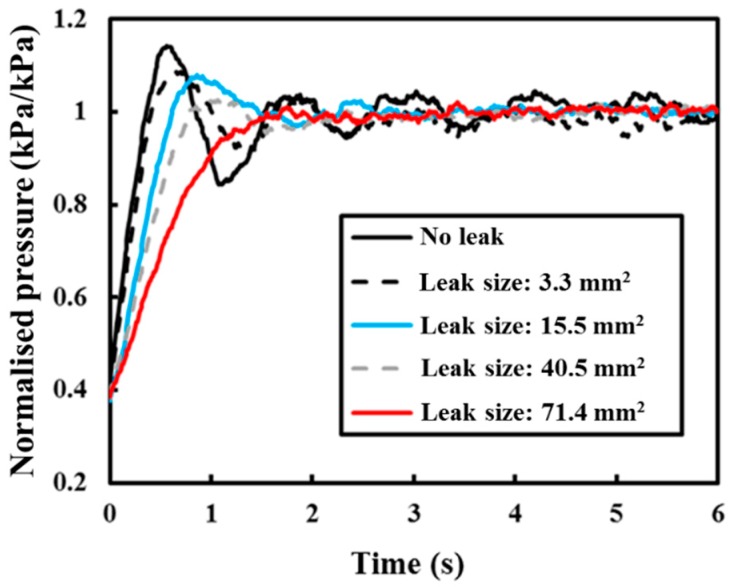
Normalised pressure profile with a different leak size.

**Figure 13 sensors-18-04192-f013:**
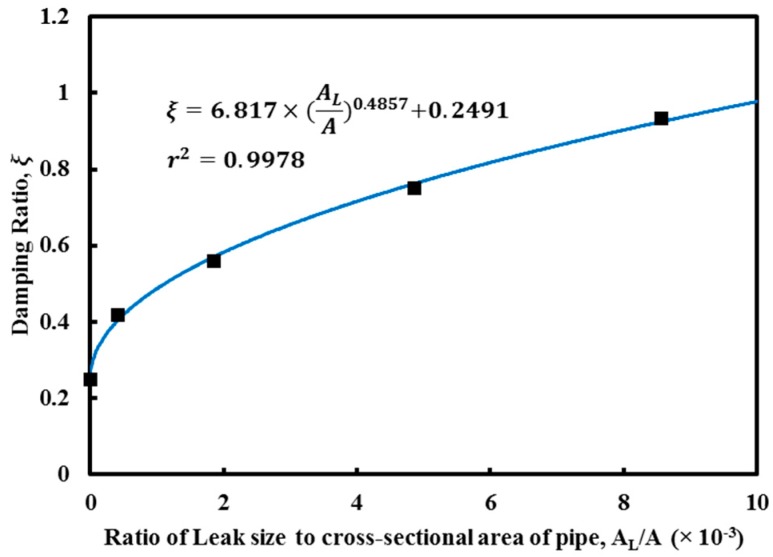
Damping ratios for different leak sizes.

**Figure 14 sensors-18-04192-f014:**
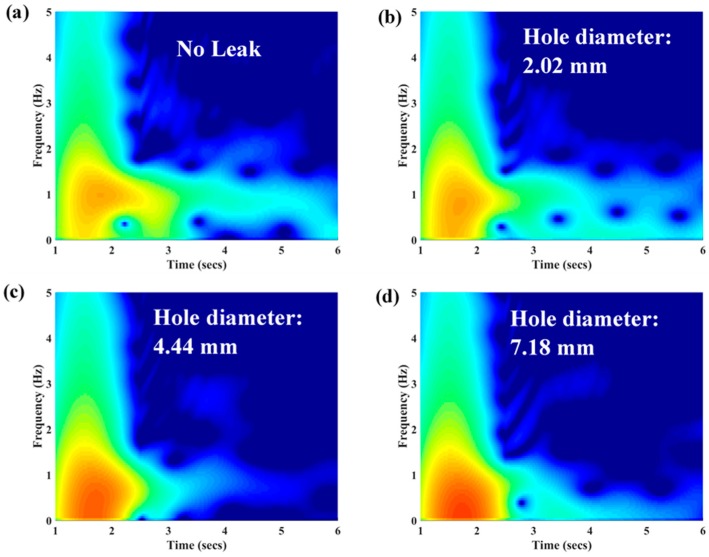
(**a**–**e**): Spectrogram of each of the results obtained with a different leak size, as shown in Figure 12; and (**f**): The relationship between the frequencies with maximum magnitude against different leak size.

**Figure 15 sensors-18-04192-f015:**
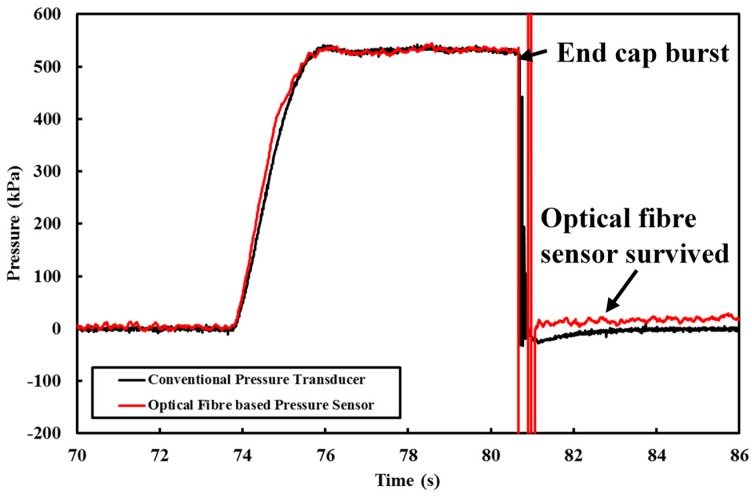
Pressure profile when the pipe burst event occurred.

**Table 1 sensors-18-04192-t001:** Specifications of the optical distributed sensor interrogator (ODiSI-B) under the standard mode of operation.

Type of Information	Description
Maximum sensing length	10 m
Maximum data acquisition rate	100 Hz
Gauge length	5.12 mm
Strain resolution	1 µε
Strain repeatability	±5 µε
Temperature resolution	0.1 °C
Temperature repeatability	±0.4 °C

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
