# Peer review of "Leak Detection in Water Pipes Using Submersible Optical Optic-Based Pressure Sensor"

_sensors, 2018, doi:10.3390/s18124192_

Round 1
Reviewer 1 Report
The paper by Wong et al. deals with a very interesting and current topic, and it has merit. Nonetheless, the paper has some issues that the Authors should carefully address before publication. For the Authors' convenience, here are these issues:
- The introduction and general background are quite misleading, as the Authors refer to the sensor technology as distributed but it is not the case. Although the Authors use an OFDR device to interrogate the sensor, the sensor is a single point sensor, compatible with multiplexing. In particular, they used an ODISiB device, that has a very short range: this device does not fit at all the needs for pipeline monitoring. However, about this, it is not the first single-point pressure sensor, where an OFDR has been used for interrogation, as in
Schenato, et al. "Fiber optic sensor for hydrostatic pressure and temperature measurement in riverbanks monitoring," Optics & Laser Technology, Volume 82, 2016, Pages 57-62, ISSN 0030-3992, https://doi.org/10.1016/j.optlastec.2016.02.015.
As stated in that paper, the Authors here should mention that the inner working mechanism of the sensor does not rely on the specific optical technique used to measure the strain (and the temperature), given that this technique has sufficient spatial resolution to measure the strain in those portions of fiber. For example, two FBGs may be conveniently used, and in that case, the range may be extended over several kilometers.
- about the state of the art, the Authors are completely ignoring DTS (Distributed Temperature sensors) system (for buried pipeline) and DAS (Distributed Acoustic Sensors) systems.
- about the calibration of the sensor: the Authors reported the strain measured by the ODisi device, but they did not clearly state where that value of strain has been calculated. ODisi has a centimetric-level spatial resolution, and therefore I imagine that some measurement points are collected within the portion of fiber glued to the PVC tube. Is the strain value reported the average over those points (as done in the aforementioned paper of Schenato et al.)? Is that value the one calculated in the middle point? It may be useful to have a curve of the strain measured along the fiber for a given pressure.
-about figure 2(a): the arrows suggest that the sensor mechanism is based on the deflection of the inner tube section, but, looking at the measurements it is not the case. In case of a deflection, the fibe undergoes to elongation, and the strain should be positive. Instead, I suspect that the mechanism is barely the contraction of the entire structure (and therefore a shortening of the fiber glued to the wall) due to the pressure. Please comment on that.
-about calibration: the Authors should provide an estimation of the accuracy of the sensor (in term of standard deviation of the error, for example).
- section 4.2.1: the degree symbol (°) is wrongly typed (o).
- fig. 9: "sec" is not the symbol for seconds, but "s"
- About static leak detection: in fig. 9, it seems that the leak test 1 at 10° (6mL/s) can be hardly identified, whereas in fig 10(a) it seems that the measured the pressure drop is not that negligible. Can the Authors explain how they calculate the pressure drop from the data? What is used as a reference and as a final pressure? E.g. the mean pressure before drop calculated over n-measurements, the pressure just before and after the drop...
- in the conclusions, the Authors mention the non-linearity of the PVC along with the hysteresis, creep and temperature dependency. Did they observe these phenomena during these tests? Did they observe any drift in the pressure response over time?
- It would be wonderful if they could provide the temperature calibration to complete the work. If not, they should clearly state that it is mandatory for the sensor to be used in practical applications.
Author Response
Hi, Thank you so much for reviewing our paper. Please find our responses in the attached documents.
Thanks

Reviewer 2 Report
The authors proposed an interesting work in the leakage detection field. The work is well structured and complete. Nevertheless, some aspects should be modified:
1. Include a space between the number and the unit. For instance, lines 133, 135, 149, 161, …
2. Pressure resolution: 4.46 kPa. Please comment this value (it is enough for the desired applications, it should be a higher value, compare with the resolution of the traditional systems).
3. Figure 6: Include a comment for the numbers 1-7, or alternatively, change the legend. For instance: “…was subjected to various changes (identified with the number 1-7) in pipe operating condition.”
4. Figure 7: Why the Action 7 was not analyzed in detail?
5. Figure 8b): Why is the “Case 3” included in the figure?
6. The English should be revised in some sentences.
This work could be published after minor revisions.
Author Response
Hi, Thank you so much for reviewing our manuscript. Please find the attached for our responses.

Round 2
Reviewer 1 Report
The Authors have amended the paper accordingly to Revieewer's requests and, in my opinion, the paper is now almost acceptable for publication: there is one point that the Authors should amend further. The sensor can be syrely deployed in daisy-chain configuration and interleaved with other sensors or approaches (DTS, DAS...), but, in order to be interrogated it requires an optical technique with enough spatial resolution to measure the strain within the 50 mm of fiber section that is bonded to the PVC tube. Given a specific pressure, a lower spatial resolution will result in a lower measured strain, as it will be averaged over a longer length (and for sure the maximum value could not be grasped, as the Authors did in this paper). And as far as I know, there is no-long range distributed (or quasi-distributed) optical technique able to guarantee such a spatial resolution, but only FBG. Therefore, the Authors shouldn't conclude that "However, it does not restricted the potential of multiplexing the same sensor to form an array of sensors.", unless they employ FBG technology.
Furthermore, I recommend the Authros to double-check the references (Authors's spelling name and doi).
Author Response
Thank you for your time again. Please find the response on the attached file.
Thanks
